# Local Scour Near Flexible Flow Deflectors

**Liquan Xie [1] , Yehui Zhu [1], Huang Li [1], Yan-hong Li [2],\* , Yuanping Yang [3] and Tsung-chow Su [4]**

[1] College of Civil Engineering, Tongji University, Shanghai 200092, China; xie_liquan@tongji.edu.cn (L.X.); yehui_zhu@tongji.edu.cn (Y.Z.); 1630446@tongji.edu.cn (H.L.)

[2] State Key Laboratory of Ocean Engineering, School of Naval Architecture, Ocean and Civil Engineering, Shanghai Jiaotong University, Shanghai 200240, China

[3] Zhejiang Institute of Hydraulics and Estuary, Hangzhou 310020, China; yangyp@zjwater.gov.cn

[4] Department of Ocean and Mechanical Engineering, Florida Atlantic University, Boca Raton, FL 33431, USA; su@fau.edu

\* Correspondence: yyhli@sjtu.edu.cn

**Abstract:** Rigid flow deflectors are usually used on water flow beds to protect engineering structures such as breakwater in coasts and to regulate flow routes in open channels. To reduce its side-effects, i.e., local scour at the toe of deflectors, a flexible flow deflector is proposed, and the corresponding local scour was investigated in this study. A flume experiment was conducted to investigate local scour. To show the advantage of flexible deflectors, a control experimental test was also conducted using a traditional rigid deflector under the same blockage area configuration and the same flow conditions. The flow field near the flexible deflector was also measured to reveal the local flow field. The results show that the bed-scour develops near the toe edges of both flexible and rigid deflectors, but the maximum and averaged scour depths for the flexible deflector are smaller. This advantage of flexible deflector in scour depth is mainly caused by its prone posture, which induces the upward stretching and enlarging horizontally rotating vortex and the upward shifted vertically rotating vortex. The former dissipates more turbulent energy and the latter results in smaller bed shear stress, which lead to smaller scour depth directly. In addition, the up- and down-swaying movement of the flexible deflector can also assistant to dissipate more turbulent energy, thereby damping the intense of the horseshoe vortices and thus weakening scour depth as well. The results of this study provide an elementary understanding on the mechanisms of a flexible flow structure and an alternative deflector-device to reduce scour depth.

**Keywords:** flexible flow deflector; scour depth; horseshoe vortex

## 1. Introduction

Underwater engineering structures, such as breakwater and cylinder in coastal areas [1–3], are usually exposed to the toe-scour damage and instability risk. To protect these underwater structures, rigid flow deflectors are widely used in the upstream section of these engineering structures. The working mechanism of flow deflector is to modify and optimize local flow pattern, leading part of the flow over the protected structures, and then reducing the scour damage risk to them. The rigid flow deflector is also used to regulate flow routes in channels and in fish habitat restoration areas. However, due to obstacle effect, horseshoe vortices are usually induced at the leeside of underwater structures and causes local scour at its toe part [4]. This local scour is also considered as a side-effect of deflectors, because it may trigger structural failure of the deflector by itself.

The morphology dynamics, including scour and deposition at the toe of submerged rigid engineering structures, such as engineering deflectors, have attracted substantial research. Most of this research has focused on the developing process of local scour and the local flow pattern [5–8], the scour depth and the scour volume [9–11], the temporal development of scour hole [12], and the sediment deposition features [13].

To protect the underwater rigid engineering structures, including deflectors, from scour-caused failure, traditional measures generally include ripraps [14,15], flexible mattresses [16], rubble-mound structures [17], permeable structures [18,19], and protective spur dikes [20]. However, these methods have their drawbacks. Ripraps and mattresses cover a large area near the underwater structure, which may bring negative effects to local ecology [21]. The rubble-mound structure is sometimes inapplicable in the circumstance of a limited space.

Besides rigid structures, flexible structures have been widely applied underwater in river management projects. If properly designed, the local scour near the flexible structures is less significant compared with that of rigid structures. Such structures include inflatable flexible membrane dams [22], mattress curtain sets [23], fiber reinforced mats [24], suspended curtains [25], geotextile mattress with sloping curtain [26,27], geotextile mattress with sloping plate [28], etc.

In this study, a novel flexible flow deflector was proposed to minimize the side-effects of local scour when modifying flow patterns. A flume experiment was conducted to investigate the scour depth at its toe. A traditional rigid flow deflector was conducted as a control experimental test. The flow field was also measured for analysis.

## 2. Flexible Flow Deflector

A flexible flow deflector is proposed, which is an underwater structure for flow pattern modification (Figure 1a,b). It is composed of a bottom beam, a flexible curtain, and a floating tube. The bottom beam serves as a foundation for the whole structure and can be made of reinforced concrete, geotextile tube, or other heavy materials. The flexible curtain is a geotextile sheet that is attached to the beam on the bottom edge and the floating tube is attached on the upper edge. The floating tube is a light or inflatable structure manufactured with materials like foamed plastic. When the flexible flow deflector is deployed in a steady current, the upper edge of the curtain is extended upward and inclines to the downstream side due to the buoyancy of the floating tube and the flow drag force. Therefore, the flexible deflector partially blocks the cross section of channel in the lower part of flow. Thus, the approaching flow is diverted to the lateral sides, or climbs over the top of deflector. At the leeside of the deflector, a three-dimensional horseshoe vortices system forms due to the blockage effect of the deflector (Figure 1c).

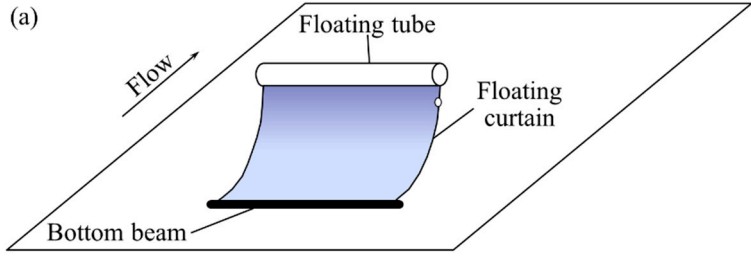

**Figure 1.** *Cont.*

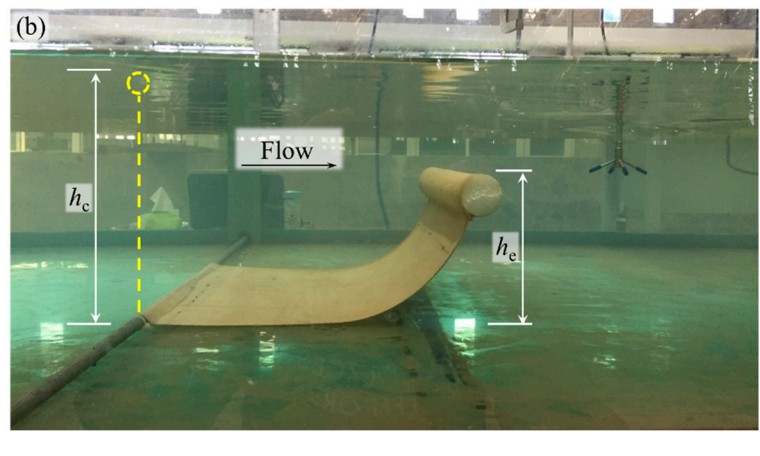

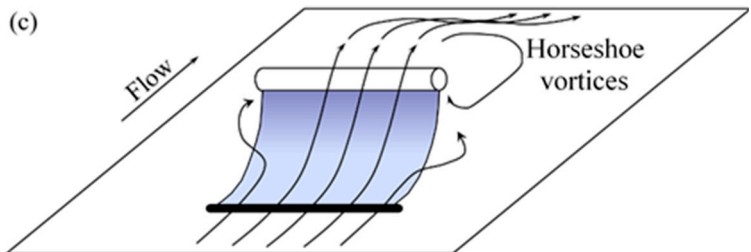

**Figure 1.** Flexible flow deflector: (**a**) Sketch of flexible flow deflector, (**b**) Photo of flexible flow deflector, (**c**) Sketch of the flow pattern near flexible flow deflector.

## 3. Experimental Setup

### 3.1. Experiments of Local Scour near Deflectors

The experiments on local scour were conducted in a water flume at Zhejiang institute of hydraulics and estuary, China. The glass-sided flume is 42 m in length, 4.3 m in width, and 1.0 m in depth (Figure 2). A honeycomb structure was installed at the entrance of the flume to stabilize the current. The maximum flow depth in the flume is 0.6 m, and the maximum flow rate is 0.8 m$^3$ s$^{-1}$. A sediment recess of 3 m in length and 0.15 m in depth was located in the center part of the flume. Four outer-sides and the inner-bottom of the recess, and the bottom of the flume were covered with impermeable concrete. Natural sand was taken from the Qiantang river estuary site and applied in the test. The median diameter of sand $d_{50}$ = 0.34 mm and the geometric standard deviation $\sigma_g = (d_{84}/d_{16})^{1/2}$ = 1.48. The velocity in the flume was monitored with an acoustic doppler velocimeter (ADV).

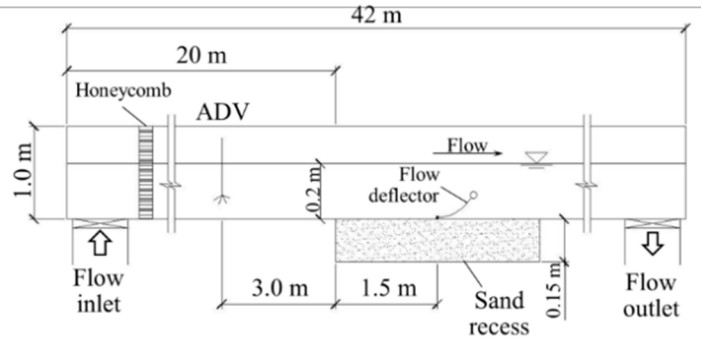

**Figure 2.** Side view of the experimental water flume (not to scale).

For the purpose of comparison, a control test was conducted with a rigid deflector under the same flow condition, the same shape configuration and the same wide and effective height as that of the flexible deflector. The rigid deflector model was a 0.4 m wide and 0.2 m high plastic plate. Half of the plate was buried in the sediment, and thus the effective blockage area of the rigid model deflector was 0.4 m in width ($W$) by 0.1 m in height ($h_c$) (Figure 3). The flexible model was composed of a bottom beam, a flexible curtain, and a floating tube. The bottom beam was a 0.4 m long steel bar with a diameter of 10 mm, being fixed on the bed with short pillars. The width of the flexible curtain of polyester fiber was $W = 0.4$ m. The original height (i.e., in full stretched condition) of the flexible curtain was $h_c = 0.3$ m. Under the test condition of flow velocity of 0.35 m s$^{-1}$ ($V$) and water depth of 0.2 m ($h$), the effective height of the flexible current $h_e = 0.1$ m (Figure 3). Thus, the widths and effective heights of the rigid and flexible models were identical. The single deflector model was located at the center of the recess in each experimental test (Figure 2).

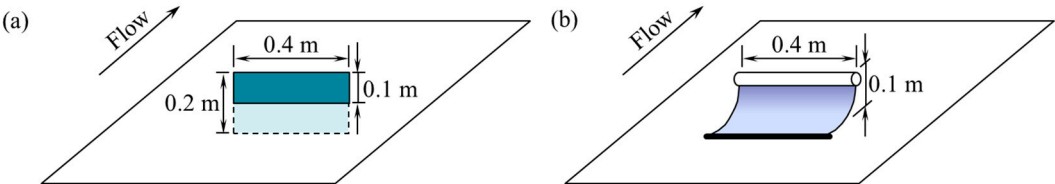

**Figure 3.** Sketch of two models of flow deflector (not to scale): (**a**) Rigid flow deflector, (**b**) Flexible flow deflector.

A point morphology-measurement gauge was carried on a sliding bar to observe the surface variation of flow bed. The accuracy of the point gauge is 0.1 cm. The measurement area covers from 0.2 m upstream of the deflector to 0.4 m downstream of the deflector. In the cross-sectional direction, the measurement area covers 0.4 m from the center line of the flume on two sides. The interval of the neighboring measurement points was 0.1 m in streamwise direction and 0.05 m in cross-sectional direction.

In this study, two comparative cases were performed to reveal the difference in scour pattern between the rigid and flexible flow deflectors. The flow velocity was 0.35 m s$^{-1}$ in the tests, the flow depth was 0.2 m, and the flow rate was 0.301 m$^3$ s$^{-1}$. Thus, the Froude number is 0.25. The steady flow was maintained for 2 h during the entire experimental test. At the end of each test, the flow was slowed down steadily so that the bed topography remained intact. The Froude number and the sediment size and composition all fall in the range of that of prototype at site. Therefore, the Froude's law of similarity can be used at site application.

### 3.2. Experiments of Flow Field near a Flexible Flow Deflector

The experiments on the flow field near a flexible flow deflector were also performed to analyze the test results of local scour. The tests were conducted in the flume where the local scour tests were conducted. Sediment transportation was not considered in the flow field tests. The velocity was measured with a down-looking ADV and a side-looking ADV. The former was used at measuring points which were more than 5 cm below the surface of the water, and the latter was used at measuring points close to the water surface. Both ADVs have the identical accuracy and configuration parameters. The full scale of the ADVs was ±1.0 m s$^{-1}$, and the accuracy was ±0.5% of measured value ±1 mm s$^{-1}$. The sampling frequency was 100 Hz. For each measuring point, the ADV readings were taken for 60 s.

The average flow velocity was 0.35 m s$^{-1}$, and the flow depth was 0.2 m, which were identical to those used in the scour test. The flexible flow deflector model used in the scour test was also used in this experiment. The detailed parameters of the deflector can be referred in the previous section.

Figure 4 shows the locations of the flow velocity measuring points, where the $x$ axis is in streamwise direction; the $y$ axis is in the lateral direction, and the $z$ axis is in vertical direction. Because the flexible

deflector was placed in the cross-sectional middle part of the flume, and the flexible deflector has no significant spanwise sway due to the stable effect of two rigid beams at its bottom and up ends, the time-averaged flow fields and the scour pattern at the two spanwise sides to the cross-sectional center of the flexible deflector will be systematic. Therefore, only a half of the spanwise of the flow field and the scour pattern in the horizontal plane was observed.

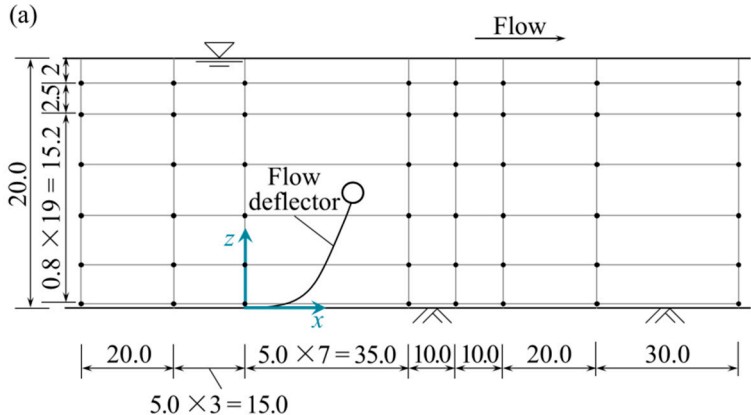

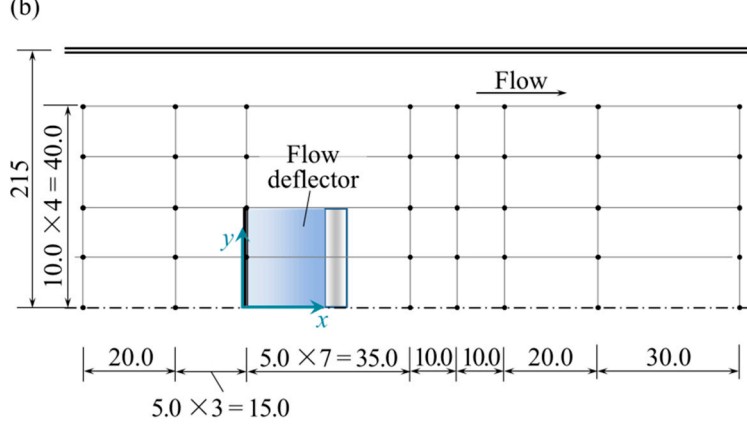

**Figure 4.** Measuring points near a flexible flow deflector (not to scale, unit: cm): (**a**) Side view, (**b**) Top view.

The flow fields were measured in vertical and horizontal planes, respectively. Three horizontal planes for flow fields were set at $z = 0.019$ m, 0.099 m, and 0.18 m, respectively. One vertical plane for flow field was set along the streamwise center line of the flume. In the horizontal plane, 40 measurement points were set (Figure 4b). In the vertical plane, the velocity was measured at 21 points along the vertical direction. It should be noted that some of the measuring points under the deflector were not measured due to the blockage of the flexible curtain.

## 4. Scour near Flexible and Rigid Flow Deflectors

Figure 5 shows the scour pattern near the rigid and flexible flow deflectors, where the bed elevation $S$ is normalized as $S/h_e$. Remarkable scour appeared at the edge of both flow deflectors (Figure 5). The scour depth, scour area, and scour volume were compared between the rigid and flexible cases. Compared with the control experimental test of the rigid deflector, which has a maximum scour depth of 8.2 cm, the maximum scour depth in the flexible deflector test is 7.0 cm, which decreases by 14%. The average scour depth decreases from 2.3 cm in rigid deflector test to 1.8 cm in the flexible deflector test, which decreases by 22%. The detailed scour information is given in Table 1.

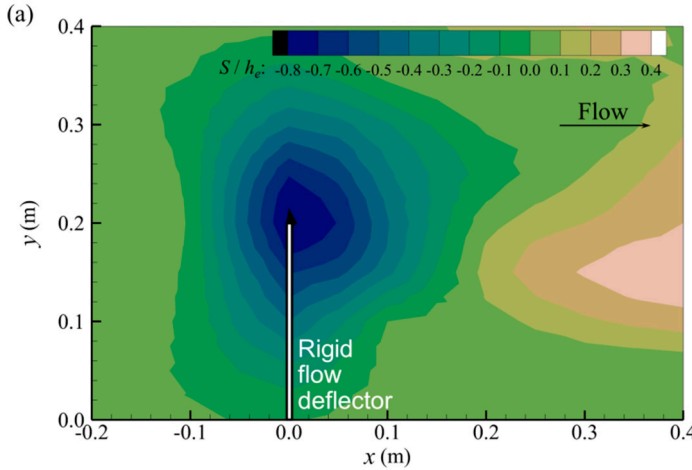

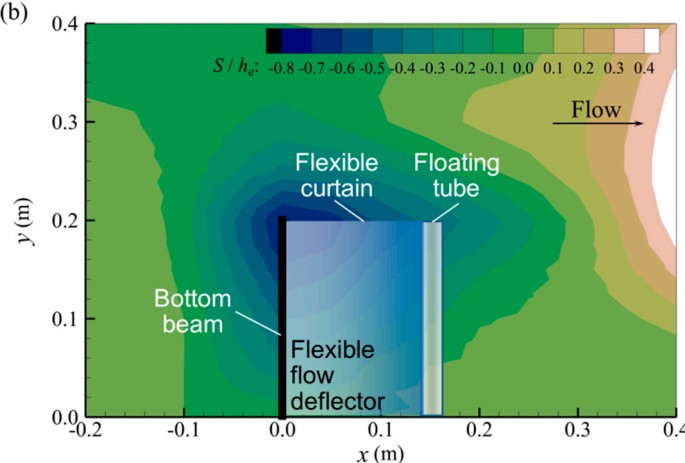

**Figure 5.** Bed topography near rigid and flexible flow deflectors (top view): (**a**) Rigid flow deflector, (**b**) Flexible flow deflector.

**Table 1.** Detailed information of the scour near the flow deflectors.

| Structure | Maximum Scour Depth (cm) | Average Scour Depth (cm) | Scour Area (cm$^2$) | Scour Volume (cm$^3$) |
|---|---|---|---|---|
| Rigid | 8.2 | 2.3 | $0.9 \times 10^3$ | $2.1 \times 10^3$ |
| Flexible | 7.0 | 1.8 | $1.1 \times 10^3$ | $2.0 \times 10^3$ |

However, the coverage of the scour hole extends further downstream than that of the rigid deflector. In the flexible case, the scour area is larger than that of the rigid case by 22%. As a result, the difference of the total scoured volume between these two tests is not so remarkable as that of scour depth. The scour volume for the flexible deflector test is less than that for the rigid deflector test, though only by 4.8%.

Compared with rigid deflector, the flexible deflector has two remarkable features: it is prone toward downstream and has an up and down swaying movement in flow. Although the effective heights of deflectors for these two cases are the same, the prone posture of flexible deflector leads the overflow to be more streamlined. Thus, the horseshoe vortices were weaker, which induces smaller scour depth. In addition, the swinging movement of flexible deflector (Figure 6) dissipates more TKE, thereby damping the intensity of horseshoe vortices and resulting in smaller scour depth.

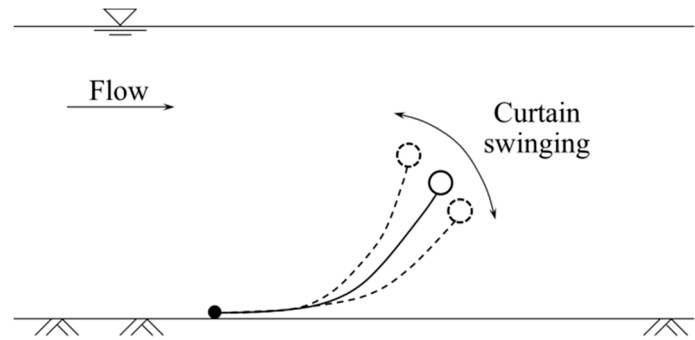

**Figure 6.** Up and down-swaying deflector in flow.

Although the flexible deflector produces a larger shelter area on the flow bed for the shedding vortices to generate, the shedding vortices have a much weaker intensity than that of the horseshoe vortices. Therefore, although the flexible deflector causes a larger scour area, the scour depth is smaller, and this benefits the stability and structural safety of the deflector. In this study, the approaching flow was clear water, and thus a slight difference may lie in the equilibrium scour depth and scour development between this study and studies with different sediment transport dynamic conditions.

## 5. Flow Field near Flexible Flow Deflector

### 5.1. Flow Velocity near a Flexible Flow Deflector

Figure 7 shows the flow fields near a flexible deflector in $x - y$ plane at different $z$ levels. The velocity vectors were normalized with the average flow velocity $V = 0.35$ m s$^{-1}$. The flow fields at different $z$ levels indicate that the presence of the flexible flow deflector causes significant variations in the flow field. Figure 7a shows the flow field near the water surface at $z = 0.18$ m where the horizontal plane is 0.08 m above the top of the flexible deflector, Figure 7b shows the flow field near the top of the flexible deflector at $z = 0.099$ m, and Figure 7c shows the flow field near the bottom end of the flexible deflector at $z = 0.019$ m. The comparison of these three figures indicates that the most deformed velocity vector occurs near the end of the bottom beam of the flexible deflector at $x = 0$ m and $y = 0.2$ m at $z = 0.019$ m (Figure 7c), and then shifts downstream with the increase of height level, occurring at $x = 0.14$ m and $y = 0.2$ m at the upper height level $z = 0.099$ m (Figure 7b), and further downstream at $x = 0.16$ m and $y = 0.2$ m at the well-above deflector height level $z = 0.18$ m (Figure 7a).

Figure 7b,c also show that a horizontally rotating vortex was observed between $x = 0.2$ m and 0.6 m. As the deflector and the approaching flow was supposed to be axisymmetric, two vertical vortices formed immediately downstream to the deflector, one clockwise and one anticlockwise. The vortices were induced by the bypass flow on two sides of the deflector and the low-pressure zone on the leeside of the deflector.

A comparison of Figure 7b,c indicates that with the increase of height level $z$, the horizontally rotating horseshoe vortex at the leeside of the flexible deflector is stretched larger in horizontal size, with its core zone shifting downstream. It can be reasonably assumed that due to the downstream-prone posture of the flexible deflector, a vertically upward stretching and enlarging vortex is formed at the leeside of the flexible deflector. The conceptual scheme of this stretching and enlarging vortex is shown in Figure 8. As a result of the upward stretch and enlarge of the horizontal rotating vortex, less scour depth will be caused, due to more dissipation of turbulent energy.

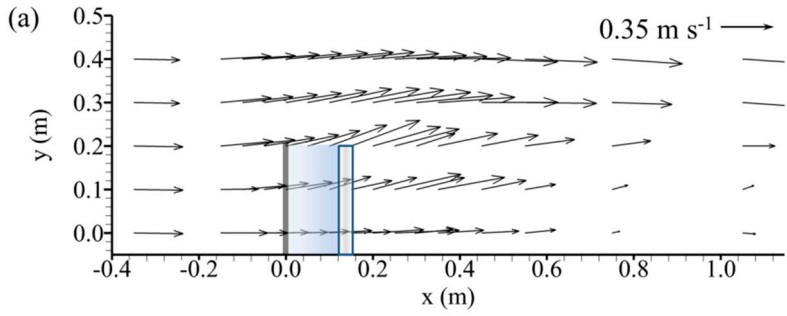

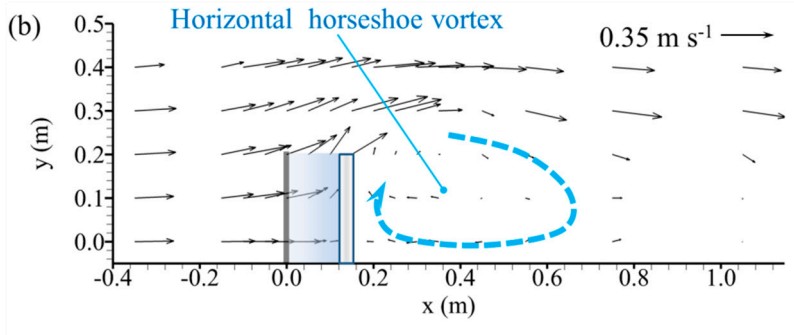

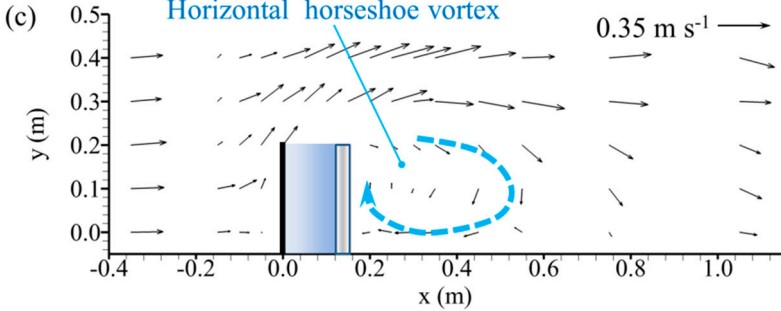

**Figure 7.** Velocity vectors at different $x - y$ planes: (**a**) $z = 0.180$ m, (**b**) $z = 0.099$ m, (**c**) $z = 0.019$ m.

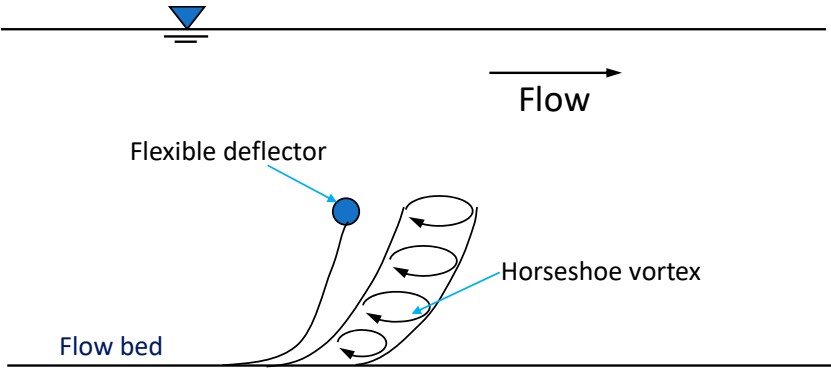

**Figure 8.** Scheme of the upward stretching and enlarging horizontally rotating vortex.

In addition, the horizontal planar extension of the scour area can also be partially attributed to these two vortexes. As the flexible deflector inclines downstream, the vortex also extends more to the leeside (Figure 7). Thus, its influencing area expands, and the area of the scoured zone enlarges.

Figure 9 shows the flow field in the vertical plane, which was observed along the streamwise center line of the flume. On the upstream side of the deflector, the approaching flow was diverted upward and accelerated remarkably. This acceleration is consistent with that observed in Figure 7a, which was attributed to the blockage effect of the deflector.

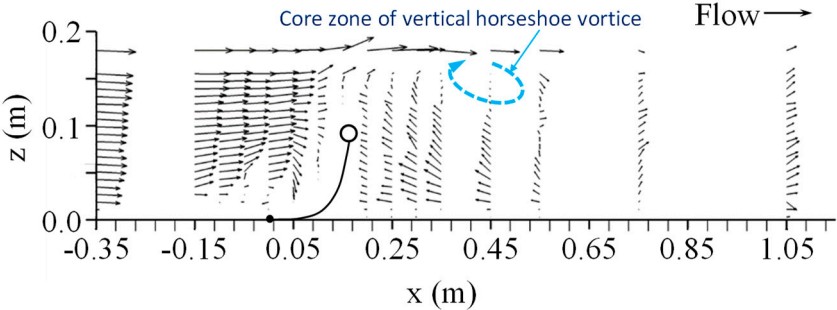

**Figure 9.** Velocity vectors on streamwise center line.

Figure 9 also shows that a vertically rotating horseshoe vortex located 1 to 4 times of the effective deflector height downstream of the up-floating-beam of the deflector. For rigid submerged structure, it is usually observed that the core zone of the horseshoe vortex is near the bottom of structure (e.g., Dey 2008 [4]). However, for this flexible deflector, a remarkable feature of the vertically rotating horseshoe vortex is that its core zone is shifted upward to the top region of the flexible deflector. At the near-bed region at the leeside of the flexible deflector, the flow velocity is significantly smaller than that of a rigid structure. The small velocity will generate small shear stress near the bed and cause a smaller scour depth.

*5.2. Turbulence Intensity near a Flexible Flow Deflector*

Figure 10 shows the vertical turbulence intensity distribution along the streamwise centerline of the flume. The relative turbulence distribution $N$ is calculated as follows. In Figure 10, the vertical axis is normalized with the flow depth $h$ as relative depth $z/h$.

The instantaneous flow velocity $u$ can be described as [29]

$$u_i = \overline{u_i} + u_i'$$

(1)

where $\overline{u}$ = time averaged velocity and $u'$ = random increment. The components of the turbulence intensity $\sigma_i$ and relative turbulence intensity are calculated as

$$\sigma_i = \sqrt{\overline{u_i'^2}}$$

(2)

$$N_i = \frac{\sigma_i}{\overline{u_*}}$$

(3)

where $u_i'$ is the component of the perturbation of flow velocity, and $\overline{u_i}$ is the component of the time averaged flow velocity.

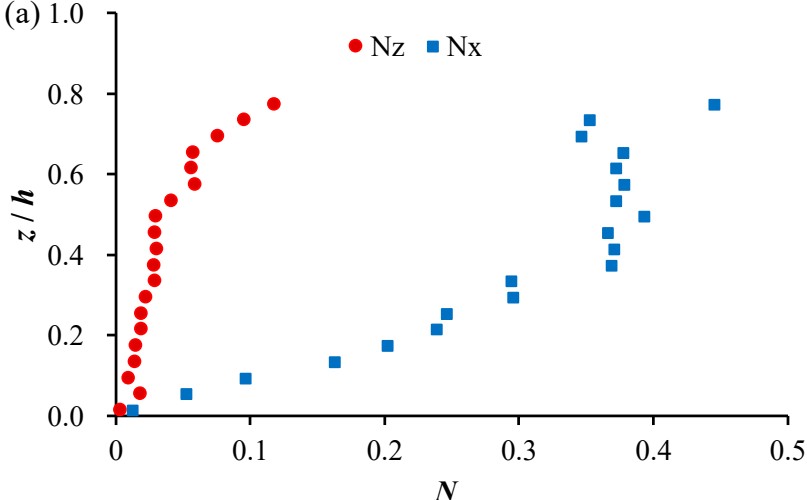

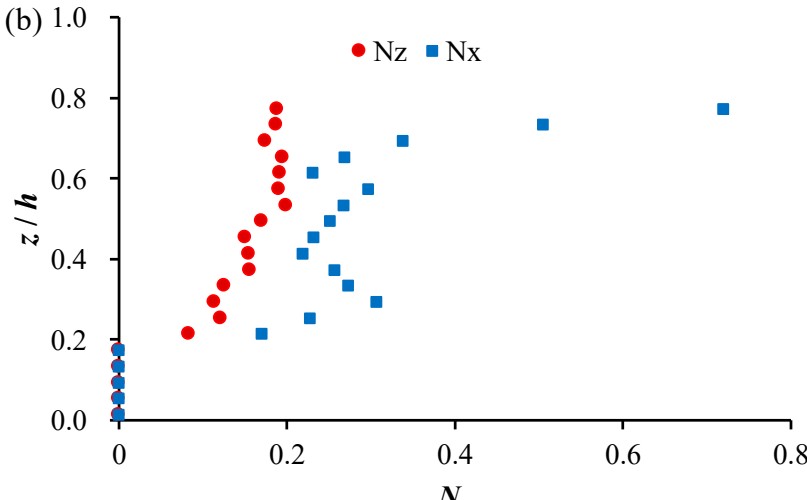

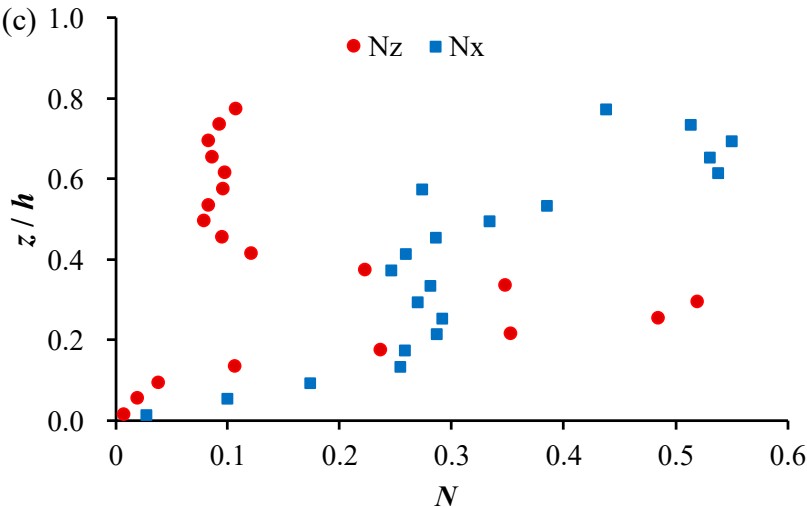

**Figure 10.** *Cont.*

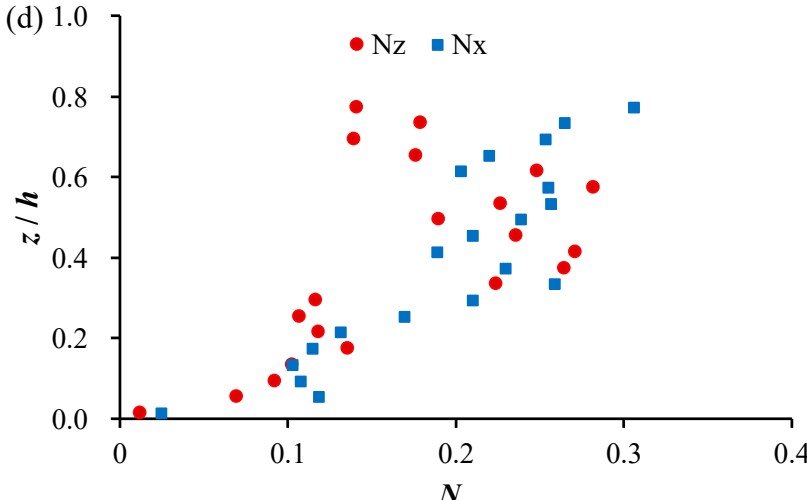

**Figure 10.** Vertical distribution of turbulence intensity along the streamwise center line: (**a**) $x = -0.35$ m, (**b**) $x = 0.05$ m, (**c**) $x = 0.25$ m, (**d**) $x = 0.75$ m.

Figure 10a–d indicates that compared with that which was upstream of the flexible deflector, the turbulence intensities near the water surface were increased at the closest leeside of the flexible deflector at $x = 0.05$ m (Figure 10b), which is caused by the deformation of the velocity vectors; further downstream at $x = 0.25$ m (Figure 10c), the turbulence intensities along the entire water depth increase due to the deformed velocity vectors; and at the far leeside of the flexible deflector at $x = 0.75$ m, where the influence of the deflector on the flow field is negligible small, the turbulence intensities decrease (Figure 10d), which suggests that the flexible deflector enhances the dissipation of the turbulent energy.

## 6. Discussion

In addition to the two main mechanisms of the upward stretching and enlarging horizontally rotating vortex and the upward shifted vertically rotating vortex at the leeside of the flexible deflector, which causes smaller scour depth, there are some other features of flexible deflector which may cause smaller scour depth, as follows:

(1) The prone posture of the flexible deflector is of a curve-pattern, which leads to a more complex flow pattern and a smaller scour depth. When deployed in a steady flow, the flexible deflector is prone downstream along an up-concave rolled curve line. At the bottom-beam part, the curtain is more significantly prone, and almost touches the flow bed (Figure 1b,c). Thus, in the upstream side of the curtain, the flow partially curves back toward the upstream direction and forms small vortices (Figure 9). This back-curved flow dissipates a part of TKE, which can help to reduce the scour depth at the leeside of curtain. A similar phenomenon has been reported in [30].

(2) In addition, the up- and down-swaying motion of the flexible deflector will assistant to dissipate more TKE, which also leads to smaller scour depth. The frequency of the up- and down-swaying of the deflector was observed in the range of 0.9–1.1 Hz, with the swaying amplitude falling in the range of 0.001–0.003 m.

(3) It should be pointed out that the scour mechanism at the toe part of the flexible flow deflector is complicated. It even involves the interactions of seepage scour. The tests and analyses in this study proposed an elementary and qualitative understanding of the horseshoe scour. To further reveal all the details and mechanisms of these interactions, more investigations need to be conducted in the future.

## 7. Conclusions

A new flexible flow deflector is proposed as a device to protect the underwater engineering structure, and the local scour depth at its leeside was investigated in this study. As a control experiment, a rigid deflector of the same effective height and width was also conducted under the same flow condition. The results show that the local scour occurs at the toe part of both rigid and flexible flow deflectors. Compared with the rigid deflector, the flexible deflector induces smaller maximum scour depth and smaller average scour depth, which enhances its stability and the structural safe performance. The smaller scour depth mainly benefited from the prone posture, which induces the upward stretching and enlarging horizontally rotating vortex and the upward shifted vertically rotating vortex. The former dissipates more turbulent energy and the latter results in smaller bed shear stress, which led directly to smaller scour depth. Besides, the up- and down-swaying movement of the flexible deflector can also assist in dissipating more turbulent energy, which causes a smaller scour depth as well.

**Author Contributions:** Conceptualization, Y.L. and L.X.; methodology, L.X. and T.S.; validation, Y.Z.; formal analysis, H.L. and T.S.; investigation, Y.Y.; resources, data curation, Y.Y.; writing original draft preparation, Y.Z.; visualization, Y.Z. and Y.Y.; supervision L.X.; project administration, Y.L. and L.X.; funding acquisition, Y.L. and L.X. All authors have read and agreed to the published version of the manuscript.

**Funding:** This research was funded by the National Natural Science Foundation of China, grant numbers 11172213 and 51479109.

**Acknowledgments:** This work is partly supported by the China Scholarship Council (grant number 201806260166).

**Conflicts of Interest:** The authors declare no conflict of interest.

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
