# Peer review of "Local Scour Near Flexible Flow Deflectors"

_water, doi:10.3390/w12010153_

Round 1

Reviewer 1 Report

See attached file.

Reviewer 2 Report

Dear Authors,

thank you for an interesting manuscript showing a laboratory experiment on a viable technical solution to be used in river engineering practice. The manuscript cannot be accepted in the present form as it lacks of novelty and is basically showing only one flow combination (water discharge, flow depth etc.). The several times mentioned and discussed flow patterns are not measured and/or presented in the manuscript. It is a clear-water experiment, no sediment is flowing into the section, you must discussed this issue with regard to which extent a sediment transport might change the local scour dynamics and dimensions for flexible and rigid flow deflectors. Nothing has been said about the model scale, even though you are using sand fractions of a given size. Different sand mixtures might influence the local scour by armouring the scour hole for example. Even though the problems seems simple it needs much more research efforts, even for a technical note. Especially in the context of flow patterns and different vortices causing local scour. You should have visualized flow by using dye or other devices to estimate turbulence kinetic energy dissipation.

Please, see full papers in this topic, such as:

https://www.mdpi.com/2073-4441/10/6/680

https://www.mdpi.com/2073-4441/11/8/1530/htm

https://www.mdpi.com/2073-4441/11/8/1580

and you will see that much is missing in your experimental set up and experimental design to be able to bring something new to the audience of the journal Water.

Reviewer 3 Report

The technical note provided by Liquan Xie , Yehui Zhu , Huang Li , Yan-hong Li  , Yuanping Yang , Tsung-chow Su is a physical model study on flow deflector which supposed to protect hydro-engineering structures from erosion caused by river flow.

Authors preceded experiment by using one flow condition and two versions of the flow deflector. Authors compared qualitatively the obtained results. It is relatively simple experiment and very little information is provided in the manuscript.

I not think the paper can be publish In present form.

My general comments are as follows:

Since the deflectors will work in natural conditions I expect the experiments to include variable flow conditions and sediment size. It should cover the conditions obeserved in real location where the prototype is planned to be install. Also it will allow calibrating the results by comparing with scours observed in reality.   Authors do not provide any information about scaling their results to the prototype. For physical model study it should be the first issue to be discussed. I mean how the water discharge and velocity, share stress, sediment size, flow resistance and turbulence intensity obtained in the model is scaled to reproduce the prototype? Flow pattern data are missed. I wonder if that was even measured, but if not experiments need to be repeated and water velocity filed should be analyzed together with the turbulence intensity – please see similar works with use of ADV data such as:
Kolerski, T., & Wielgat, P. (2014). Velocity Field Characteristics at the Inlet to a Pipe Culvert. Archives of Hydro-Engineering and Environmental Mechanics61(3-4), 127-140. The paper size is limited, thus it should be paid more attention to filter the information and show only these which are important. For example figure 1 can be reduced to only (a) and (b) and show it next to each other. Consider if table 1 is needed – all information are in text already. I not sure if MDPI Water require ‘Notation’ section, please remove if not necessary. Data provided in the text are not consistent with the sketch on figure 2 – flow depth, ADV location.

Round 2

Reviewer 2 Report

Dear Authors,

thank you for the clarification on the majority of comments and suggestions, especially for adding the details about the flow field. Please, specify also the used discharge during your experiments: the max. is 0.8 m3/s, and your applied was 4.3m x 0.2m x 0.35 m/s = 0.301 m3/s. maybe in line 104, where you have introduced Froude number Fr=0.25.

There are still some grammatical correction needed:

line 117 - "field" instead of "filed"

line 117 - "deflector" instead of "defletor"

line 175 - sediment "transport" dynamics conditions

line 204 - Figure 7 needs a scale for flow velocity vectors, please, show the length of V = 0.35 m/s

line 222 - Figure 9 needs a scale for velocity vectors!

line 215 - please, discuss the position of the horseshoe vortex in dimensionless units (instead of 0.18-0.78 m), use the flow depths and/or height of the deflector instead (e.g. 1 to 4 times the flow depth downstream of the deflector reaching up to half of the flow depth)

line 224 - You are referring to Figure 9, it should be Figure 10

line 225 - change Figure 9 to Figure 10

line 237 - change figure caption from Figure 9 to Figure 10

lines 239-244 - change Figure 9 to Figure 10

line 258 - you are mentioning swaying motion of the flexible deflector - since you have measured by ADV flow velocities for 60 s, could you give an average frequency of this swaying motion of the deflector, just to give an insight into this dynamics compared to flow measurements

Reviewer 3 Report

The revised version of the manuscript is significantly improved. I appreciate authors comments and additional work done to the paper. I will suggest some references for Equations 1 - 3 

it could be the same as in other authors' paper: 

Kolerski, T. and Wielgat, P., 2014. Velocity Field Characteristics at the Inlet to a Pipe Culvert. Archives of Hydro-Engineering and Environmental Mechanics61(3-4), pp.127-140.

or other reference
